# On Explaining Equivariant Graph Networks via Improved Relevance Propagation

## Abstract

We consider explainability in equivariant graph neural networks for 3D geometric graphs. While many XAI methods have been developed for analyzing graph neural networks, they predominantly target 2D graph structures. The complex nature of 3D data and the sophisticated architectures of equivariant GNNs present unique challenges. Current XAI techniques either struggle to adapt to equivariant GNNs or fail to effectively handle positional data and evaluate the significance of geometric features adequately. To address these challenges, we introduce a novel method, known as EquiGX, which uses the Deep Taylor decomposition framework to extend the layer-wise relevance propagation rules tailored for spherical equivariant GNNs. Our approach decomposes prediction scores and back-propagates the relevance scores through each layer to the input space. Our decomposition rules provide a detailed explanation of each layer's contribution to the network's predictions, thereby enhancing our understanding of how geometric and positional data influence the model's outputs. Through experiments on both synthetic and real-world datasets, our method demonstrates its capability to identify critical geometric structures and outperform alternative baselines. These results indicate that our method provides significantly enhanced explanations for equivariant GNNs.

## 1 Introduction

Equivariant graph neural networks have shown significant promise in addressing complex problems across quantum physics, molecular science, materials science, and protein research (Thomas et al., 2018; Fuchs et al., 2020; Liao & Smidt, 2022; Liao et al., 2023; Batzner et al., 2022; Passaro & Zitnick, 2023; Zhang et al., 2023; Yu et al., 2023; Du et al., 2024). Despite their potential, a critical challenge in assessing the scientific plausibility of these models' outcomes is their interpretability. Most equivariant GNNs are treated as black boxes, which undermines their reliability and limits their applicability in scientific domains. Therefore, developing explainable artificial intelligence (XAI) techniques tailored for equivariant GNNs is highly desirable. These techniques can provide insights into how equivariant GNNs make predictions, thereby increasing the trustworthiness of their outcomes. Moreover, XAI techniques can not only diagnose and improve existing models but also facilitate further scientific knowledge discovery.

While many XAI methods have been proposed to study GNNs, they primarily focus on 2D graphs (Yuan et al., 2023; 2020; Zheng et al., 2023; Chen et al., 2024; Wang et al., 2021). The high dimensionality of 3D geometric data and the complexity of equivariant GNN models pose unique challenges and opportunities in this domain. Current XAI techniques either struggle to adapt to equivariant GNNs or fail to effectively handle positional data and evaluate the significance of geometric features adequately. Specifically, many XAI methods (Huang et al., 2022; Zhang et al., 2021; Vu & Thai, 2020) overlook the complex behavior of equivariant models, thus requiring additional effort before they can be applied to equivariant GNNs. On the other hand, some XAI methods, known for their simplicity and adaptability, such as SA (Baldassarre & Azizpour, 2019), are insufficient to provide a comprehensive explanation for the importance of geometric features.

To fill this gap, we introduce a novel XAI method called EquiGX, which measures the importance of input components by decomposing the model predictions. The primary challenge in decomposing the predictions of spherical equivariant GNNs lies in attributing the tensor product-based message-passing operations that are central to these networks. Our approach uses the Deep Taylor

decomposition framework to extend layer-wise relevance propagation rules specifically for spherical equivariant GNNs. By explicitly considering the tensor product (TP) operations, we derive new relevance propagation rules based on Taylor decomposition. These rules enable us to back-propagate relevance scores layer by layer until the input space, providing a detailed explanation of each layer's contribution to the network's predictions. Consequently, EquiGX can enhance our understanding of how geometric and positional data influence the model's outputs.

## 2 BACKGROUND AND RELATED WORK

We denote a geometric graph with $n$ nodes as $\mathcal{G} = \{\mathbf{X}, \mathbf{A}, \mathbf{C}\}$. Here, $\mathbf{X} = [\mathbf{X}_1, \cdots, \mathbf{X}_n]^T \in \mathbb{R}^{n \times d}$ is the node feature matrix, where each $\mathbf{X}_i \in \mathbb{R}^d$ is the $d$-dimensional feature vector of node $i$. $\mathbf{C} = [\mathbf{C}_1, \cdots, \mathbf{C}_n]^T \in \mathbb{R}^{n \times 3}$ is the node coordinate matrix, where $\mathbf{C}_i$ is the coordinate of $i$-th node. Nodes are generally connected by edges using a predetermined radial cutoff distance $c \in \mathbb{R}^+$, so that the adjacency matrix $\mathbf{A} \in \{0, 1\}^{n \times n}$ is defined as $\mathbf{A}_{ij} = 1$ if and only if $\|\mathbf{C}_i - \mathbf{C}_j\|_2 \leq c$. We use $\mathcal{N}(i)$ to denote the set of neighboring nodes of node $i$.

### 2.1 EQUIVARIANT GRAPH NETWORKS

Equivariant graph neural networks are critical in the domain of AI for science, particularly for modeling geometric graphs derived from three-dimensional atomic systems. These networks are specifically designed to capitalize the physical symmetries and integrate these symmetries into the model architecture to ensure that the learned hidden representations are equivariant to any symmetry transformations applied to the input. Specifically, if the input geometric graph is transformed under any operation in $SE(3)$, which stands for the special Euclidean group in 3D space, the corresponding hidden representations at each layer are transformed correspondingly. Formally, a function $f : \mathbb{R}^{n \times 3} \rightarrow \mathbb{R}^{2\ell+1}$ mapping between 3D coordinates to a $(2\ell + 1)$-dimensional vector is $SE(3)$ equivariant, if for any input coordinates $\mathbf{C}$, we have $f(\mathbf{R}\mathbf{C}^T + \mathbf{t}) = D^\ell(\mathbf{R})f(\mathbf{C})$, where $\mathbf{t} \in \mathbb{R}^3$ is a translation vector, $\mathbf{R}$ is a rotation matrix satisfying $\mathbf{R}^T\mathbf{R} = \mathbf{I}$ and $|\mathbf{R}| = 1$, and $D^\ell(\mathbf{R}) \in \mathbb{R}^{(2\ell+1) \times (2\ell+1)}$ represents the Wigner-D matrix of $\mathbf{R}$ (Gilmore, 2008). Here, function $f$ is invariant to translation, exemplifying a specific type of translation equivariance.

Among the various types of equivariant GNNs (Jing et al., 2020; Schütt et al., 2021; Satorras et al., 2021), spherical equivariant GNNs (Thomas et al., 2018; Fuchs et al., 2020; Liao & Smidt, 2022) are particularly prominent. In these approaches, spherical harmonics functions are used to first encode 3D geometric information into higher dimensional $SE(3)$ equivariant features. We denote the order-$\ell_1$ $SE(3)$ equivariant hidden features of node $i$ as $H_i^{\ell_1} \in \mathbb{R}^{2\ell_1+1}$. These features are used in a tensor product operation to compute an equivariant message from node $i$ to node $j$, denoted as $M_{j \rightarrow i}$, and the aggregated message $M_i = \sum_{j \in \mathcal{N}(i)} M_{j \rightarrow i}$ is used to update the equivariant hidden features. $M_{j \rightarrow i}$ consists of many features with multiple rotation orders as $M_{j \rightarrow i} = \bigoplus_{\ell=0}^{\ell_{max}} M_{j \rightarrow i}^\ell$, where $\bigoplus$ is direct sum. For an order-$\ell_3$ message $M_{j \rightarrow i}^{\ell_3}$, it can be computed by using the order-$\ell_2$ spherical harmonics function as

$$M_{j \rightarrow i}^{\ell_3} = \sum_{\ell_1, \ell_2} \mathbf{F}^{(\ell_1, \ell_2, \ell_3)}(d_{ij}) \mathbf{Y}^{\ell_2}(\vec{r}_{ij}) \otimes H_j^{\ell_1}. \tag{1}$$

Here, $\mathbf{F}(\cdot)$ is a learnable function usually implemented by a multi-layer perceptron (MLP) model, $d_{ij} = \|\mathbf{C}_i - \mathbf{C}_j\|_2$ and $\vec{r}_{ij} = \frac{\mathbf{C}_i - \mathbf{C}_j}{d_{ij}}$ are the distance and direction between nodes $i$ and $j$, respectively. $\mathbf{Y}^{\ell_2}(\cdot) : \mathbb{R}^3 \rightarrow \mathbb{R}^{2\ell_2+1}$ is the spherical harmonics function, which maps an input 3D vector to a $(2\ell_2 + 1)$-dimensional vector representing the coefficients of order-$\ell_2$ spherical harmonics bases. $\otimes$ is the tensor product operation, which takes a order-$\ell_1$ equivariant feature $\mathbf{u}$ and a order-$\ell_2$ equivariant feature $\mathbf{v}$ as input, yielding order-$\ell_3$ equivariant feature as

$$(\mathbf{u}^{\ell_1} \otimes \mathbf{v}^{\ell_2})_{m_3}^{\ell_3} = \sum_{m_1=-\ell_1}^{\ell_1} \sum_{m_2=-\ell_2}^{\ell_2} \mathcal{C}_{(\ell_1, m_1),(\ell_2, m_1)}^{(\ell_3, m_3)} \mathbf{u}_{m_1}^{\ell_1} \mathbf{v}_{m_2}^{\ell_2}, \tag{2}$$

where $\mathcal{C}$ is Clebsch-Gordan (CG) coefficients (Griffiths & Schroeter, 2018) and $m$ denotes the $m$-th element in the equivariant feature. See more discussions about equivariant graph neural networks in Appendix D.

## 2.2 Explainability in Graph Neural Networks

Explainability in neural networks is vital for validating the trustworthiness and reliability of their predictions, especially when applying these models to scientific domains. Current XAI methods predominantly focus on GNNs designed for 2D graphs. These approaches can be mainly categorized into four classes, namely, gradients/feature-based methods, perturbation-based methods, decomposition methods, and surrogate methods. Gradients/Feature-based methods, such as SA (Baldassarre & Azizpour, 2019) and CAM (Pope et al., 2019), use gradient values to assess the importance of input components. Their popularity stems from their simplicity and direct approach. Perturbation-based methods (Ying et al., 2019; Yuan et al., 2021; Luo et al., 2020) evaluate changes in predictions by perturbing different input features to identify the most impactful ones. Surrogate-based methods (Huang et al., 2022; Zhang et al., 2021; Vu & Thai, 2020) involve fitting a simpler, interpretable model, such as a decision tree, to mimic the behavior of the original model. The surrogate model's explanations are then used to understand the original predictions. Decomposition methods (Schnake et al., 2021; Xiong et al., 2023; Feng et al., 2023) decompose prediction scores and back-propagate them layer-by-layer to the input space to compute importance scores and provide deeper insights into each network layer. Despite significant advances in XAI for 2D GNNs, these methods primarily focus on evaluating the importance of edges, nodes, and subgraphs, struggling to incorporate positional information effectively and fully evaluate the importance of geometric features. Consequently, the application of these techniques to 3D geometric graphs, especially within equivariant graph neural networks, poses significant challenges. Recently, Miao et al. (2023) introduces a learnable interpreter model that applies random noise to each 3D point to generate importance scores. However, this method treats the models as black boxes and overlooks the equivariance of the model. It also requires training the interpreter alongside the prediction model. To sum up, the challenge of explaining equivariant neural networks highlights a significant gap in the current landscape of XAI, underscoring the need for innovative approaches that consider the complex behaviors of equivariant neural networks.

## 3 Methodology

Previous XAI methods on 2D graphs encounter limitations when adapting them on geometric graphs, particularly in effectively incorporating positional information and evaluating geometric features. To address these challenges, we introduce a novel method, EquiGX, which recursively decomposes network predictions back to the input elements. Our approach use the Deep Taylor decomposition framework (Montavon et al., 2017), adapted to extend the layer-wise relevance propagation rules specifically for TP based message passing process. This adaptation allows for a detailed explanation of each layer's contribution to the network's predictions, thus enhancing our understanding of how geometric and positional data influence the model's outputs.

### 3.1 Layer-wise Relevance Propagation

The objective of Layer-wise Relevance Propagation (LRP) is to attribute a relevance score to each input element based on its contribution to the predicted class. This scoring offers insights into how individual input elements contribute to the model's final decision. One way to compute such relevance is to the whole neural network as a mathematical function and use the first-order term from the Taylor series expansion. Consider a function $f : \mathcal{X} \to \mathcal{Y}$ that maps an input to its output label. The Taylor decomposition of $f$ at a root point $\hat{\mathbf{x}} \in \mathbb{R}^d$ is given by

$$f(\mathbf{x}) = f(\hat{\mathbf{x}}) + \sum_i \frac{\partial f}{\partial x_i}\bigg|_{\mathbf{x}=\hat{\mathbf{x}}} (x_i - \hat{x}_i) + \mathcal{O}(|\mathbf{x} - \hat{\mathbf{x}}|^2), \tag{3}$$

where $\mathcal{O}$ is Big-O notation, and $x_i$ and $\hat{x}_i$ is the $i$-th dimension of $\mathbf{x}$ and $\hat{\mathbf{x}}$, respectively. Assuming $f$ is a locally linear function and carefully selecting $\hat{\mathbf{x}}$ such that higher-order and zero-order terms are negligible, the first-order terms can provide the relevance scores for the input elements as $\mathcal{R}(x_i) = \frac{\partial f}{\partial x_i}\big|_{\mathbf{x}=\hat{\mathbf{x}}} (x_i - \hat{x}_i)$. Deep neural networks are inherently complex and non-linear, making it impractical to apply a straightforward Taylor decomposition across all layers. On the other hand, Deep neural networks, composed of multiple layers, necessitate decomposing the network into a series of simpler subfunctions, each representing a single layer. This approach, known as Deep

Taylor Decomposition, allows for applying different relevance score computation rules tailored to specific types of layers. For instance, when considering linear layer with Relu activation functions, distinct rules, such as LRP-$\gamma$ (Montavon et al., 2019), LRP-$\alpha\beta$ (Bach et al., 2015) can be used due to choosing different root points and approximation methods. By using these specifically designed local propagation rules for every layer, the initial relevance value, i.e. the prediction of the model, is successively distributed layer-by-layer to the input space. The decomposition characteristic of LRP gives rise to the conservation property, which ensures that the sum of relevance scores across neurons in two adjacent layers remains constant. Let $H$ and $H'$ be the representations of two adjacent layers, the conservation property can be formally described as $\sum_i \mathcal{R}(H) = \sum_j \mathcal{R}(H')$, where $\mathcal{R}(H)$ and $\mathcal{R}(H')$ are the relevance scores of $H$ and $H'$, respectively. We use the Deep Taylor decomposition to study the complex behavior of equivariant GNNs and provide detailed relevance propagation rules for each layer in the following subsections.

## 3.2 ATTRIBUTING THE TP-BASED MESSAGE PASSING

As mentioned in Section 2.1, the key of spherical equivariant GNNs is the TP based message passing process. Equivariant messages $M_{j\rightarrow i}$ are computed from node $j$ to node $i$ using TP, and then aggregated to form the message $M_i$. The aggregation operation $M_{j\rightarrow i} = \sum_{j \in \mathcal{N}(i)} M_{j\rightarrow i}$ inherently provides a decomposition. Specifically, we assign a relevance score $\mathcal{R}(M_{j\rightarrow i})$ to each message proportional to its contribution to the aggregated message. Since messages of different orders are summed separately, each order is also considered individually when backpropagating the relevance score. Formally, this process can be described as $\mathcal{R}(M_{j\rightarrow i}^{\ell_3}) = \frac{M_{j\rightarrow i}^{\ell_3}}{\sum_{j \in \mathcal{N}(i)} M_{j\rightarrow i}^{\ell_3}} \mathcal{R}(M_i^{\ell_3})$.

For the equivariant message shown in Eq. 1, we can apply a Taylor series expansion to derive a decomposition rule. Specifically, the first order Taylor series expansion of an order-$\ell_3$ message $M_{j\rightarrow i}^{\ell_3}$ at a root point $\hat{H}_j^{\ell_1}$ is given by

$$M_{j\rightarrow i}^{\ell_3} = \hat{M}_{j\rightarrow i}^{\ell_3} + \sum_{\ell_1, \ell_2} \left. \frac{\partial M_{j\rightarrow i}^{\ell_3}}{\partial H_j^{\ell_1}} \right|_{H_j^{\ell_1} = \hat{H}_j^{\ell_1}} (H_j^{\ell_1} - \hat{H}_j^{\ell_1}), \tag{4}$$

where $\frac{\partial M_{j\rightarrow i}^{\ell_3}}{\partial H_j^{\ell_1}} \in \mathbb{R}^{(2\ell_3+1) \times (2\ell_1+1)}$ is a Jacobian matrix. Each element of this matrix is defined as

$$\left( \frac{\partial M_{j\rightarrow i}^{\ell_3}}{\partial H_j^{\ell_1}} \right)_{m_3, m_1} = \sum_{\ell_2} \sum_{m_2=-\ell_2}^{\ell_2} \mathrm{F}^{(\ell_1, \ell_2, \ell_3)}(d_{ij}) \mathcal{C}_{(\ell_1, m_1), (\ell_2, m_2)}^{(\ell_3, m_3)} \mathrm{Y}^{\ell_2}(\vec{r}_{ij}). \tag{5}$$

The bilinearity of the tensor product indicates that it is linear with respect to each input. This property implies that the Jacobian matrix $\frac{\partial M_{j\rightarrow i}^{\ell_3}}{\partial H_j^{\ell_1}}$ is independent of the choice of root point $\hat{H}_j^{\ell_1}$. Additionally, the absence of quadratic or higher-degree terms in the Taylor expansion suggests that when a root point is chosen such that the zero-order term equals to zero, the Taylor expansion serves as a decomposition of the message. Given that $H_j^{\ell_1}$ contributes to messages of various nodes and different orders, it is necessary to aggregate these contributions. Formally, this relevance propagation rule can be described as

$$\mathcal{R}(H_j^{\ell_1}) = \sum_{\ell_3, i} \left( \mathcal{R}(M_{j\rightarrow i}^{\ell_3}) \oslash M_{j\rightarrow i}^{\ell_3} \right)^T \frac{\partial M_{j\rightarrow i}^{\ell_3}}{\partial H_j^{\ell_1}} \odot H_j^{\ell_1} \tag{6}$$

where $\oslash$ is Hadamard division and $\odot$ is Hadamard multiplication.

However, this decomposition overlooks the contribution of relative positional information between node $i$ and node $j$. As shown in Eq. 1, spherical equivariant GNNs split the relative position vector of node $i$ and node $j$ into a distance part $d_{ij}$ and a directional part $\vec{r}_{ij}$. The directional part $\vec{r}_{ij}$ is encoded into an $SE(3)$ equivariant feature vector using spherical harmonics functions, which then serves as one input to the tensor product. The distance part $d_{ij}$ is encoded into embeddings via radial basis functions (RBF), which in turn are used to determine the weight of each tensor product path $(\ell_1, \ell_2 \rightarrow \ell_3)$. Thus, an alternative and highly desirable solution is to decompose the relevance score

of each message $M_{j \to i}$ to all three components, namely the hidden features, directional part, and distance part. Notably, the message is a trilinear function, meaning it remains linear with respect to one component when the others are held constant. Following Achtibat et al. (2024), it is reasonable to assign equal relevance values to each component. Formally, we have the relevance propagation rules as

$$\mathcal{R}(H_j^{\ell_1}) = \sum_{\ell_3, i} \left( \frac{\mathcal{R}(M_{j \to i}^{\ell_3})}{3} \oslash M_{j \to i}^{\ell_3} \right)^T \frac{\partial M_{j \to i}^{\ell_3}}{\partial H_j^{\ell_1}} \odot H_j^{\ell_1},$$

$$\mathcal{R}\left( \mathrm{F}^{(\ell_1, \ell_2, \ell_3)}(d_{ij}) \right) = \left( \frac{\mathcal{R}(M_{j \to i}^{\ell_3})}{3} \oslash M_{j \to i}^{\ell_3} \right)^T \frac{\partial M_{j \to i}^{\ell_3}}{\partial \mathrm{F}^{(\ell_1, \ell_2, \ell_3)}(d_{ij})} \odot \mathrm{F}^{(\ell_1, \ell_2, \ell_3)}(d_{ij}), \quad (7)$$

$$\mathcal{R}\left( \mathrm{Y}^{\ell_2}(\vec{r}_{ij}) \right) = \sum_{\ell_3} \left( \frac{\mathcal{R}(M_{j \to i}^{\ell_3})}{3} \oslash M_{j \to i}^{\ell_3} \right)^T \frac{\partial M_{j \to i}^{\ell_3}}{\partial \mathrm{Y}^{\ell_2}(\vec{r}_{ij})} \odot \mathrm{Y}^{\ell_2}(\vec{r}_{ij}).$$

Since one edge distance $d_{ij}$ contributes to multiple TP paths, we sum up relevance scores to get the contribution of edge's distance as $\mathcal{R}(d_{ij}) = \sum_{\ell_1, \ell_2, \ell_3} \mathcal{R}(\mathrm{F}^{(\ell_1, \ell_2, \ell_3)}(d_{ij}))$. Similarly, the direction of each edge is encoded into multiple orders of equivariant features using spherical harmonics functions, thus we sum up relevance scores to attribute the contribution of an edge's direction as $\mathcal{R}(\vec{r}_{ij}) = \sum_{\ell_2} \mathcal{R}(\mathrm{Y}^{\ell_2}(\vec{r}_{ij})))$.

Note that the relevance propagation rule discussed here is to attribute a single TP-based message passing layer. To apply relevance propagation across the entire network recursively, only the relevance score of hidden feature $\mathcal{R}(H)$ continues to backpropagate towards the input. In contrast, $\mathcal{R}(d_{ij})$ and $\mathcal{R}(\vec{r}_{ij})$ do not continue to backpropagate beyond their respective layer. These scores indicate the contributions of the edge distance and edge direction, respectively, to the final prediction within that specific message passing layer. Thus, the relevance scores $\mathcal{R}(d_{ij})$ and $\mathcal{R}(\vec{r}_{ij})$ at each message passing layer are summed to derive the cumulative relevance score for edge distances and directions.

## 3.3 ATTRIBUTING THE LINEAR OPERATION

The tensor product provides a mechanism for interactions between equivariant features of different orders, while the linear layer is designed to mix equivariant features of the same order. Specifically, this layer linearly combines each group of order-$\ell$ equivariant features to produce new features, with each group having its own set of learnable parameters. Consider the input to the linear layer as $p$ order-$\ell_1$ features of node $i$, denoted by $H_i^{\ell_1} \in \mathbb{R}^{p \times (2\ell+1)}$. The output of the linear layer is $q$ order-$\ell_1$ features of node $i$, represented as $H'_i^{\ell_1} \in \mathbb{R}^{q \times (2\ell+1)}$. Formally, the transformation in the linear layer can be described as

$$H'_i^{\ell_1} = w^{\ell_1} H_i^{\ell_1}, \quad (8)$$

where $w^{\ell_1} \in \mathbb{R}^{q \times p}$ are the learnable parameters used for mixing order-$\ell_1$ features. Since each new feature is a weighted sum of the input features, we follow the fundamental LRP-$\epsilon$ (Bach et al., 2015) to derive the propagation rule for this linear layer. Let $(H_i^{\ell_1})_{m_1}$ and $(H'_i^{\ell_1})_{m_2}$ denote the $m_1$-th and $m_2$-th order-$\ell_1$ features of node $i$ for the input and output, respectively, and let $w_{m_2, m_1}^{\ell_1}$ denote the element at the $m_2$-th row and $m_1$-th column of $w^{\ell_1}$. The propagation rule for the linear layer is defined as

$$\mathcal{R}\left( (H_i^{\ell_1})_{m_1} \right) = \sum_{m_2} \left( w_{m_2, m_1}^{\ell_1} (H_i^{\ell_1})_{m_1} \right) \oslash \left( (H'_i^{\ell_1})_{m_2} + \epsilon \mathbf{1} \right) \mathcal{R}\left( (H'_i^{\ell_1})_{m_2} \right), \quad (9)$$

where $\epsilon \in \mathbb{R}$ is a stabilizing factor with a small value, and $\mathbf{1} \in \mathbb{R}^{2\ell_1+1}$ is a all-ones vector, which broadcasts $\epsilon$ into a vector. It is worth noting that while the above relevance propagation rule is specifically for order-$\ell_1$ features of node $i$, in practice, the input contains groups of equivariant features of various orders across all nodes. Thus, the propagation rule is applied separately for every node and rotation order to compute the relevance score for all input features.

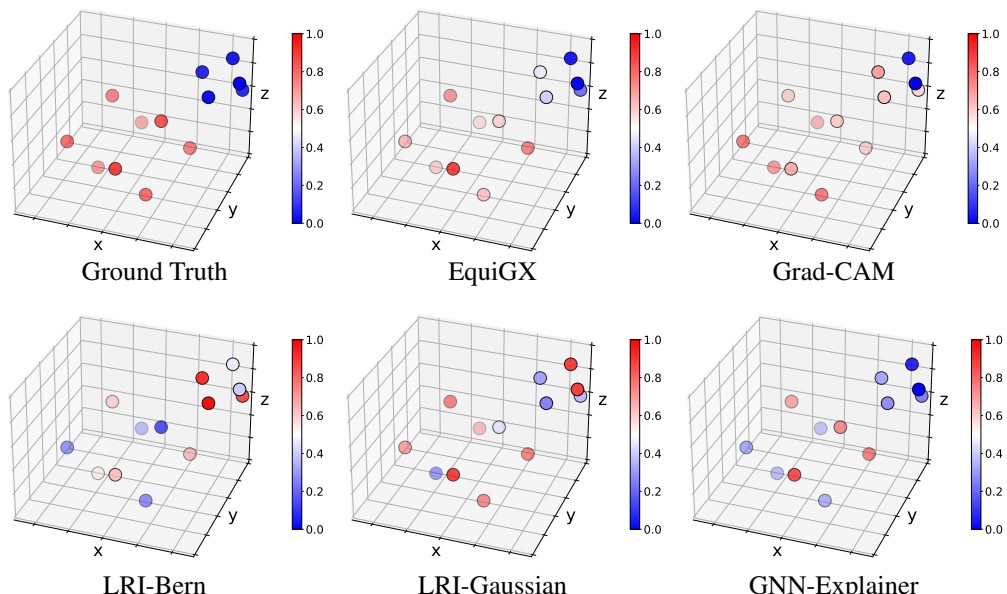

Figure 1: Explanation results on the Shapes dataset with a cube motif shape. The red color indicates a high importance score, while the blue color indicates a low importance score. Ideally, the nodes of the cube should be red, indicating their high significance, while the other areas should be blue, indicating lower significance.

### 3.4 ATTRIBUTING THE NON-LINEAR FUNCTIONS

In this work, we assume that norm-based non-linear function is used in the model architecture, such as TFN (Thomas et al., 2018) and SE(3)-Transformer (Fuchs et al., 2020). The norm-based non-linearity acts as a scalar transformation on each equivariant feature based on its norm. Specifically, for an order-$\ell_1$ equivariant feature of node $i$, denoted as $H_i^{\ell_1} \in \mathbb{R}^{(2\ell+1)}$, a scalar value is computed using an activation function like the sigmoid function. The output of this norm-based non-linearity, denoted as $H'^{\ell_1}_i \in \mathbb{R}^{(2\ell+1)}$, is computed by multiplying the input equivariant feature by the scalar output of the activation function. Formally, this process can be described as

$$H'^{\ell_1}_i = \sigma(\|H_i^{\ell_1}\|)H_i^{\ell_1}, \tag{10}$$

where $\sigma(\cdot)$ is the sigmoid function. Since each equivariant feature is transformed by a scalar, reversing the transformation results in a way to attribute relevance values. However, directly reversing the scalar transformation does not preserve the sum of relevance scores between input and output, thereby breaking the conservation property. To address this, we normalize the relevance scores to ensure the conservation property is maintained. The relevance propagation rule for the norm-based non-linear function is given by

$$\mathcal{R}(H_i^{\ell_1}) = \lambda \frac{\mathcal{R}(H'^{\ell_1}_i)}{\sigma(\|H_i\|)}, \tag{11}$$

where $\lambda \in \mathbb{R}$ is a normalization factor defined as $\lambda = \frac{\sum_{\ell_1,i} \mathcal{R}(H'^{\ell_1}_i)}{\sum_{\ell_1,i} \mathcal{R}(H'^{\ell_1}_i)/\sigma(\|H_i\|)}$.

## 4 EXPERIMENTS

In this section, we evaluate the proposed method on both synthetic and real-world datasets. For each dataset, we first train a TFN and then use baselines and our method to generate the explanations. Experimental results show that our method outperforms many baselines on both visualization results and quantitative studies. See more implemental details in Appendix A.

### 4.1 DATASETS AND EXPERIMENTAL SETTINGS

**Synthetic Datasets.** We create two kinds of geometric graph classification datasets, namely Shapes and Spiral Noise. For the Shapes dataset, we begin by randomly selecting a 3D motif shape from two options, including a cube or an icosahedron, the latter being a polyhedron with 20 triangular faces. Subsequently, we choose a 3D base shape, either a pyramid or a star. A random translation and rotation are performed on the base shape. The classification task is to predict whether the motif shape in the geometric graph is a cube or not. In the Spiral Noise dataset, we randomly select a 3D motif shape, either a tetrahedron, a polyhedron with four triangular faces, or a triangular prism. We then introduce a variable number of noise points to create a spiral pattern in 3D space. The classification task is to determine whether the motif shape is a tetrahedron or not.

**Real-world Datasets.** In addition to synthetic datasets containing perfect 3D geometric shapes, we evaluate our method on three real-world datasets, including the Structural Classification of Proteins (SCOP), BioLiP, and Actstrack. The SCOP database (Murzin et al., 1995; Andreeva et al., 2007; Chandonia et al., 2019) is a predominantly manually curated classification of protein structural domains, organized based on similarities in their structures and amino acid sequences. While using the same training and validation datasets as Hou et al. (2018); Hermosilla et al. (2020), our focus is on the fold classification task, which is to predict the broad types of protein tertiary structure topologies. Hence, we only use the Fold test set. There are seven categories in total, such as all-alpha and all-beta proteins. Protein labels, provided by human experts, are based on the secondary structure, which reflects the local spatial conformation of proteins. Specifically, labeling for all-alpha and all-beta proteins is determined by the presence of $\alpha$-helices and $\beta$-sheets within their structures, respectively. BioLiP (Yang et al., 2012; Zhang et al., 2024) is a semi-manually curated database dedicated to high-quality ligand-protein binding interactions. The 3D structural data primarily sourced from the Protein Data Bank are complemented with biological information, such as binding affinity scores, from literature and other databases. The task is to predict whether there is a tight binding between a protein-ligand pair. Like previous methods (Somnath et al., 2021; Öztürk et al., 2018; Townshend et al., 2020), we do not differentiate between the inhibition constant ($K_i$) and dissociation constant ($K_d$), instead predicting whether a protein-ligand pair is of affinity of $K_d/K_i \leq 1$ nM. ActsTrack (Miao et al., 2023) is a particle tracking simulation dataset in high-energy physics. The task is to predict whether a collision event contains a $z \rightarrow \mu\mu$ decay based on a point cloud of detector hits. Each point in the point cloud corresponds to a particle interaction with the detector. Positive samples include hits from both the $z \rightarrow \mu\mu$ decay and background interactions, thus the particle hits left by the two muons ($\mu s$) are labeled as the ground truth for model explanations.

**Baselines.** We compare our method with the following baseline methods, including (1) Grad (Baldassarre & Azizpour, 2019), which uses the norm of the gradient of the predictions with respect to the 3D coordinates to evaluate node importance; (2) Grad-CAM (Pope et al., 2019), a gradient-based method combining with activations from hidden node representations; (3) GNN-Explainer (Ying et al., 2019), a perturbation-based method identifying important edges through optimization of soft masks; (4) LRI-Bern (Miao et al., 2023), which learns a model to inject Bernoulli noise to evaluate the significance of point existence; (5) LRI-Gaussian (Miao et al., 2023), which learns a model to inject Gaussian noise to evaluate the significance of point positions; (6) PG-Explainer (Luo et al., 2020), which generate explanations by learning parameterized masks that highlight the most relevant subgraphs. For methods that assign importance scores to edges, we distribute the score to the connecting nodes to evaluate node-level explanations.

### 4.2 QUALITATIVE EVALUATION

In this section, we present the visualization of explanations for our methods and other baselines across all four datasets. Since the importance scores of different methods vary in range, we normalize each method to have the same score range to enable fair comparison. The explanation results for the Shapes dataset are visualized in Figure 1. In this dataset, the cube shape is the motif shape, so the nodes forming the cube are used as the ground truth for explanations. Therefore, the cube nodes should be marked as important, while the other nodes should not be. As shown in Figure 1, LRI incorrectly marks some nodes of the base shape as important. In contrast, our method provides better visual explanations, accurately identifying the cube nodes as the important ones. For the Spiral Noise dataset, the tetrahedron shape is the motif shape, so the nodes forming the tetrahedron are used as the ground truth for explanations. Consequently, the tetrahedron nodes should be

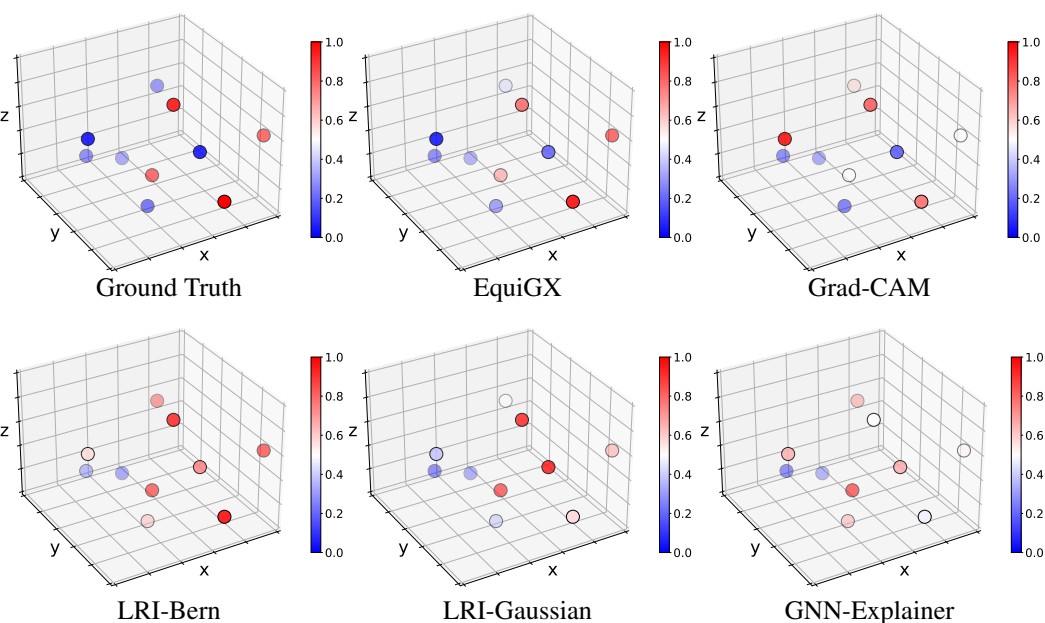

Figure 2: Explanation results on the Spiral Noise dataset with a tetrahedron motif shape. The red color indicates a high importance score, while the blue color indicates a low importance score. Ideally, the nodes of the tetrahedron should be red, indicating their high significance, while the other areas should be blue, indicating lower significance.

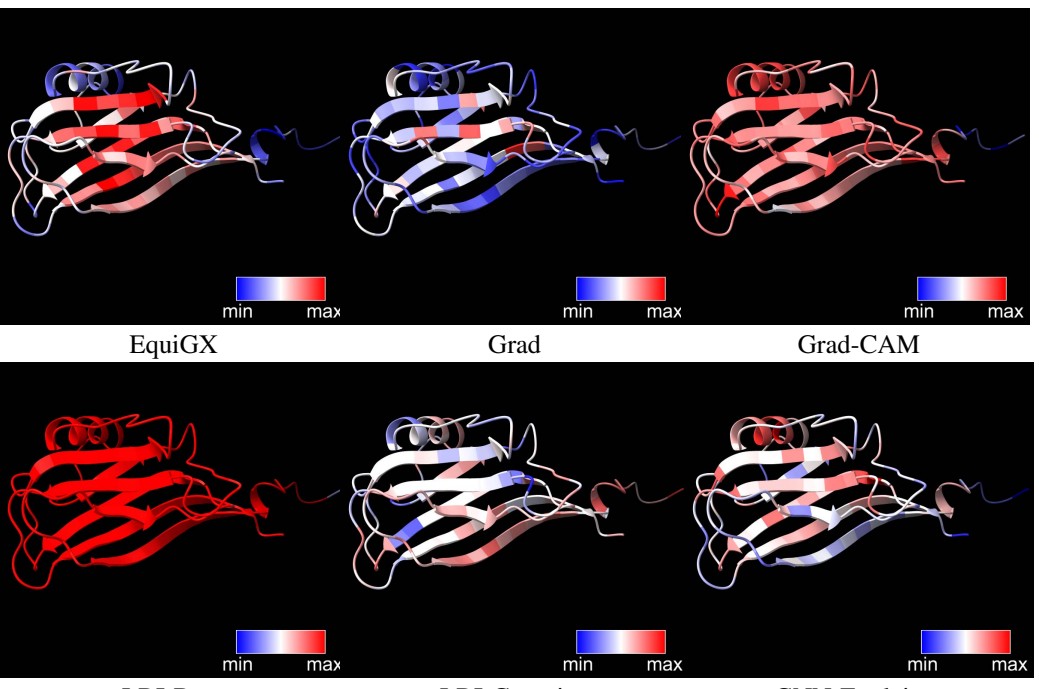

Figure 3: Explanation results on the SCOP dataset of all-beta proteins. Since the sample is an all-beta protein, ideally the $\beta$-sheets should have high importance scores, i.e. be red in the figure.

highlighted as important, while the other nodes should not be. As seen in Figure 2, GNN-Explainer struggles to identify the four important nodes forming the tetrahedron. In contrast, our method successfully recognizes the tetrahedron. We also show the explanation results of the SCOP dataset in Figure 3. As mentioned in section 4.1, protein fold classes are labeled by human experts based on

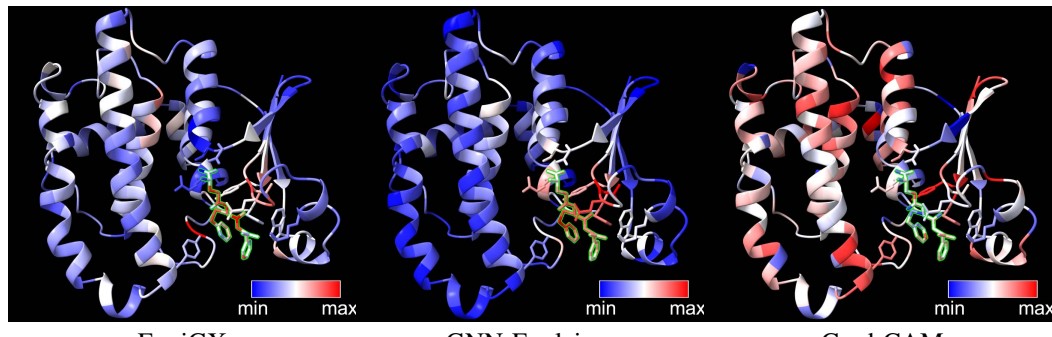

Figure 4: Explanation results on the BioLip dataset. The ligand is highlighted with a green border.

Table 1: Comparisons between our method and baselines. The best results are shown in bold.

| Dataset | Shapes | | Spiral Noise | | SCOP | | ActsTrack | |
|---|---|---|---|---|---|---|---|---|
| | AUROC ↑ | AP ↑ | AUROC ↑ | AP ↑ | AUROC ↑ | AP ↑ | AUROC ↑ | AP ↑ |
| Random | 50 | 65.70 | 50 | 49.01 | 50 | 53.67 | 50 | 20.9 |
| Grad | 68.44 ± 12.44 | 83.81 ± 6.40 | 49.94 ± 0.13 | 49.16 ± 0.09 | 56.45 ± 4.93 | 59.75 ± 3.68 | 55.84 ± 0.05 | 31.87 ± 0.54 |
| Grad-CAM | 64.77 ± 8.84 | 78.95 ± 4.82 | 66.93± 6.89 | 71.88 ± 6.45 | 59.57 ± 4.38 | 61.26 ± 1.99 | 62.11 ± 1.93 | 44.95 ± 1.40 |
| GNN-Explainer | 80.85 ± 5.38 | 89.97 ± 2.27 | 79.69 ± 2.30 | 82.37 ± 1.90 | 77.26 ± 0.19 | 72.37 ± 0.26 | 65.18 ± 0.59 | 35.54 ± 0.76 |
| LRI-Bern | 67.84 ± 17.32 | 83.25 ± 9.04 | 79.06 ± 5.69 | 81.85 ± 4.85 | 56.09 ± 2.92 | 58.45 ± 3.12 | 62.63 ± 1.43 | 39.39 ± 0.88 |
| LRI-Gaussian | 68.46 ± 10.71 | 81.65 ± 7.24 | 58.75 ± 10.96 | 63.89 ± 8.53 | 65.99 ± 5.05 | 64.35 ± 5.41 | 57.54 ± 6.21 | 32.43 ± 1.02 |
| PG-Explainer | 82.83 ± 11.7 | 90.86 ± 5.66 | 69.09 ± 1.71 | 74.53 ± 1.58 | 76.92 ± 0.23 | 72.63 ± 0.13 | 52.16 ± 4.24 | 29.43 ± 2.91 |
| EquiGX | **84.31 ± 8.89** | **91.00 ± 5.32** | **83.57 ± 10.07** | **86.82 ± 8.30** | **81.51 ± 4.61** | **82.69 ± 3.49** | **76.96 ± 1.69** | **60.47 ± 1.71** |

the secondary structures of proteins. We investigate whether the explanations provided by different methods can accurately reflect the secondary structures of proteins. An all-beta protein is shown in Figure 3. Ideally, the $\beta$-sheets should have a high importance score (i.e., be red in the figure), while the remaining parts should have a low importance score (i.e., be blue in the figure). While baseline methods either fail to identify $\beta$-sheets or incorrectly assign high importance to most parts of the protein, our method accurately distinguishes $\beta$-sheets from other parts, including an $\alpha$-helix. For the BioLip dataset, we present the explanation results in Figure 4. Since binding affinity does not have a definitive answer, there is no ground truth for explanations. It is known that binding is closely related to the protein pocket and especially the ligand itself. In the example, both our method and GNN-Explainer indicate that the model relies on the ligand to make predictions. To further evaluate explanation methods on the BioLip dataset, we conduct experiments using Fidelity and Sparsity scores in Section 4.3.

## 4.3 QUANTITATIVE EVALUATION

In two synthetic datasets, the relationships between geometric graphs and labels are explicitly defined. This allows us to evaluate the explanations of baseline methods and our approach by comparing them with the ground truth. Specifically, in the Shapes dataset, the explanation ground truth for class 0 is the nodes that form a cube, and for class 1, the nodes that form an icosahedron. Similarly, in the Spiral Noise dataset, the explanation ground truth for class 0 is the nodes that form a tetrahedron, while for class 1, it is the nodes that form a triangular prism. For both synthetic datasets, we use AUROC and average precision as evaluation metrics. As shown in Table 1, our proposed method outperforms the baselines in terms of both AUROC and average precision. In the SCOP dataset, the classification of proteins is determined based on the secondary structures of proteins. In this paper, we explain two classes, including all-alpha and all-beta proteins. Since the reason for labeling for all-alpha and all-beta proteins is the presence of $\alpha$-helices and $\beta$-sheets within their structures, respectively, we use the atoms that form $\alpha$-helices and $\beta$-sheets as the explanation ground truth. We also use AUROC and average precision as evaluation metrics. As shown in Table 1, our proposed method has better explanations than the baselines in terms of both AUROC and average precision.

For the BioLip dataset, like many other scientific properties, the rationale behind the binding affinity scores remains a topic of research itself, with no definitive answers available. Therefore, we use Fidelity and Sparsity metrics to evaluate the explanations (Pope et al., 2019; Yuan et al., 2021). The Fidelity metric assesses whether the explanations are faithfully important for the predictions by removing the identified important parts from the input geometric graphs and comparing the prediction differences. The Sparsity metric quantifies the proportion of important structures identified by the explanation methods. Note that higher Sparsity scores, which indicate that smaller structures are identified as important, can influence Fidelity scores. This is because smaller structures tend to be less crucial. The results are shown in Figure 5 where we plot the curves of Fidelity scores with respect to the Sparsity scores.

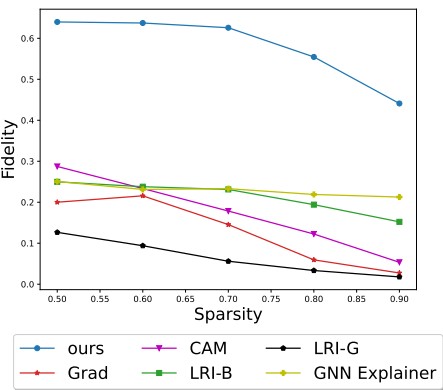

Figure 5: The quantitative studies for different explanation methods on the BioLip dataset.

Notably, the model appears not to use the binding site information for its predictions. This conclusion is supported by the low fidelity score, which remains around 0.02 when the binding sites are masked.

## 5 CONCLUSIONS

In this work, we propose a method, known as EquiGX, to explain equivariant graph neural networks for geometric graphs. Our method recursively decomposes network predictions back to the input elements. We adapts the Deep Taylor decomposition framework to TP based message passing process, leading to specifically designed layer-wise relevance propagation rules. Experimental results demonstrate the capability of EquiGX to identify critical geometric structures and provide significantly enhanced explanations for equivariant GNNs.

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

Table 2: Statistics and properties of four datasets.

| Dataset | Shapes | Sprial Noise | SCOP | BioLip |
|---------|--------|--------------|---------|--------|
| #graphs | 1000 | 1000 | 13738 | 26934 |
| #classes | 2 | 2 | 2 | 7 |
| #avg nodes | 14.92 | 10.45 | 498.49 | 320.33 |
| #avg edges | 160.94 | 89.94 | 6133.25 | 1427.3 |

Table 3: Prediction task performance of TFN models.

| Dataset | Shapes | Sprial Noise | SCOP | BioLip |
|---------|--------|--------------|---------------|------------------|
| ACC | 100 | 100 | $84.35 \pm 0.26$ | $83.66 \pm 0.89$ |
| AUROC | 100 | 100 | N/A | $83.36 \pm 0.85$ |

# A    DATASETS AND EXPERIMENTAL SETTINGS

In this section, we provide more details of our experiments. We use NVIDIA RTX A6000 GPUs for all our experiments.

## A.1    DATASETS

The statistics and properties of the datasets are reported in Table 2. For the Shapes dataset, we randomly select a 3D motif shape from two options, namely a cube or an icosahedron. The cube has a side length of 2, and the icosahedron has a radius of $\sqrt{3}$. For the base shape, we choose either a pyramid or a star. The pyramid has a base length and height of 1, while the star has an arm length of 1. A random vector is then used to translate the base shape, ensuring that it remains a certain distance from the motif shape without overlapping. Additionally, the motif shape undergoes a random rotation. The classification task is to predict whether the motif shape in the geometric graph is a cube or not. We use a radial cutoff of 5 to construct the geometric graph.

In the Spiral Noise dataset, we randomly select a 3D motif shape, either a tetrahedron or a triangular prism. The tetrahedron has a radius of 1, and the triangular prism has a length and height of 1. The chosen motif shape is transformed using a randomly sampled translation vector and rotation matrix. Next, we randomly sample 4 to 8 noise points, which form a spiral pattern with a radius of 1 in 3D space. The classification task is to determine whether the motif shape is a tetrahedron. We use a radial cutoff of 2 to construct the geometric graph.

For the SCOP dataset, we extract the backbone atoms of the protein to construct the geometric graph. Specifically, for each amino acid residue of the protein, the backbone atoms (i.e., nitrogen N, alpha carbon CA, and carbon C) are extracted and used as the nodes of the geometric graph. The atom type and residue index are used as features for each atom. We apply a radius cutoff of 5Å to create the geometric graph.

For the BioLip dataset, we extract the backbone atoms of the proteins and all atoms of the ligands to construct the geometric graph. Specifically, we use the alpha carbon CA of each amino acid residue in the protein as the nodes of the geometric graph. Additionally, every atom of the ligand is also used as a node in the graph. The atom type and residue type serve as node features. A radius cutoff of 10 Å is applied to create the geometric graph.

## A.2    TFN MODEL

We evaluate our methods and baselines using Tensor Field Network models. Each TP-based message passing layer is followed by a linear layer and a norm-based non-linear function. We first use spherical harmonics functions to compute the equivariant features of each edge up to order-$l_{max}$. These equivariant edge features are then aggregated and concatenated with the node features to produce the first hidden equivariant features. Table 4 provides details on the number of layers, the number of hidden equivariant features, and the highest order of equivariant feature $l_{max}$ in the TFN. The accuracy and AUROC of the TFN model is reported in Table 3.

Table 4: Hyperparameters for TFN models.

| Dataset | Shapes | Sprial Noise | SCOP | BioLip |
|---|---|---|---|---|
| #layers | 2 | 2 | 4 | 3 |
| #channels | 16 | 16 | 8 | 16 |
| $l_{max}$ | 3 | 3 | 3 | 2 |

Table 5: Runtime comparison between different methods.

| Inference Time | Shapes | Sprial Noise | SCOP | BioLip |
|---|---|---|---|---|
| Grad | 0.056s | 0.066s | 0.21s | 0.11s |
| Grad-CAM | 0.067s | 0.068s | 0.22s | 0.12s |
| GNN-Explainer | 0.07s | 0.058s | 0.23s | 0.1s |
| LRI-Bern | 0.13s | 0.16s | 0.35s | 0.24s |
| LRI-Gaussian | 0.15s | 0.14s | 0.33s | 0.28s |
| EquiGX | 0.2s | 0.19s | 0.36s | 0.25s |

## A.3 EVALUATION METRICS

In addition to common metrics, such as AUROC and AP, we also use Fidelity and Sparsity scores. In this section, we provide detailed definitions of these scores. Given an input geometric graph $\mathcal{G}$, XAI methods compute an importance score for each node. Based on these scores, we compute a hard node mask that contains only binary values. Using this mask, we can generate a masked graph $\mathcal{G}'$, where important nodes are masked out. Let $f$ denote a well-trained equivariant GNN. The Fidelity score is computed as

$$\text{Fidelity} = f(\mathcal{G})_y - f(\mathcal{G}')_y, \tag{12}$$

where $f(\mathcal{G})_y$ and $f(\mathcal{G}')_y$ means the predicted probability of class $y$ of graph $\mathcal{G}$ and $\mathcal{G}'$, respectively. Intuitively, Fidelity measures the change in predictions when important input elements are removed. In addition, we use Sparsity to measure the fraction of important nodes in the explanations as

$$\text{Sparsity} = 1 - \frac{|\mathcal{G}'|}{|\mathcal{G}|}, \tag{13}$$

where $|\mathcal{G}'|$ and $|\mathcal{G}'|$ denote the number of nodes in $\mathcal{G}'$ and $\mathcal{G}'$, respectively. The final Fidelity and Sparsity scores are averaged over the test dataset. Note that good explanations should exhibit high Sparsity along with high Fidelity.

## B MORE EXPLANATIONS

In this section, we show more visualizations of explanations. The explanations of the Shapes dataset are reported in Figure 6. In addition, the explanations of the Spiral dataset are reported in Figure 7. As shown in these results, our proposed EquiGX can identify the motif shapes. Furthermore, we also show explanation results of the SCOP dataset in Figure 8. An all-alpha protein is shown in Figure 8. Ideally, the $\alpha$-helices should have a high importance score (i.e., be red in the figure), while the remaining parts should have a low importance score (i.e., be blue in the figure). Our method can distinguish $\alpha$-sheets from other parts, assigning a low importance score to the remaining part. In Figure 9, we also show more explanations of our proposed EquiGX on the BioLip datasets. The results demonstrate that ligands typically exhibit high importance scores. This observation aligns with existing knowledge, which suggests that different ligands have varying binding affinity scores when interacting with the same protein.

## C RUNTIME STUDY

In this section, we conduct runtime experiments on different datasets, evaluating the runtime of each method for a single data example. It is important to note that PGExplainer requires additional training time apart from inference time. The results in the Table 5 indicate that our method has a comparable runtime to most baselines, whereas GNN-Explainer exhibits a significantly high runtime and PGExplainer incurs an additional training time cost ranging from hours to days.

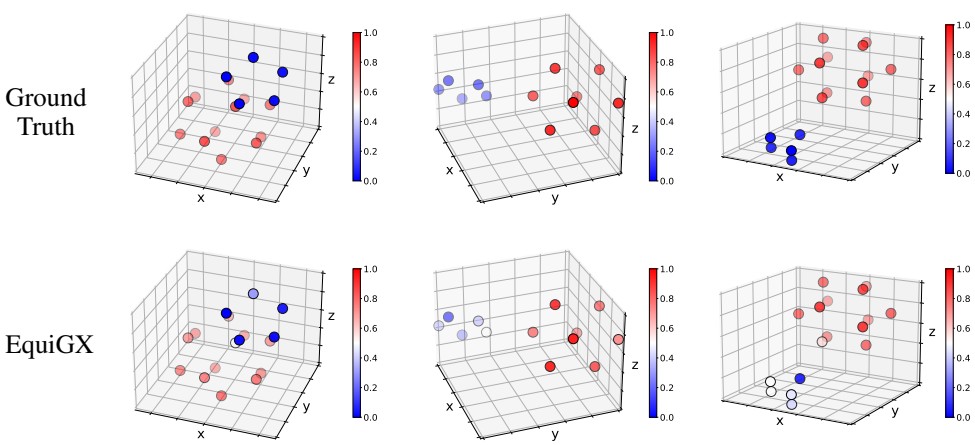

Figure 6: Explanation results on the Shapes dataset.

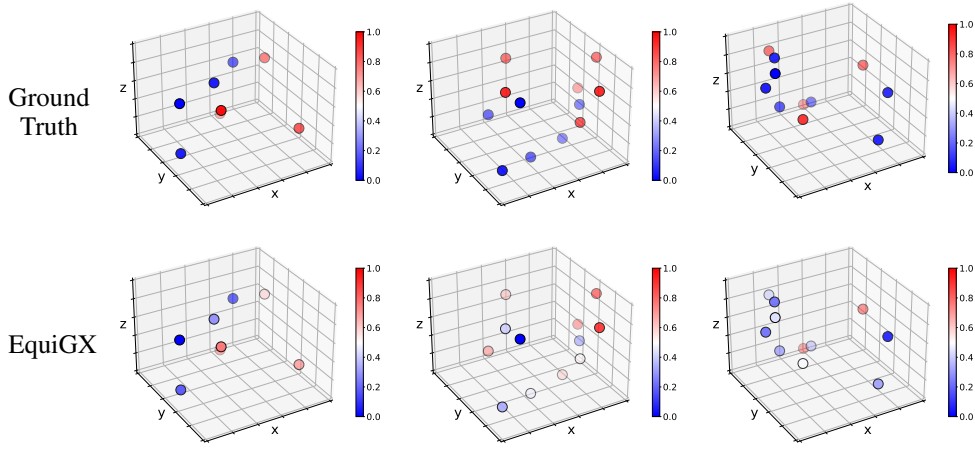

Figure 7: Explanation results on the Spiral dataset.

## D    MORE DISCUSSIONS ABOUT EQUIVARIANT GRAPH NETWORKS

As mentioned in Duval et al. (2023a), equivariant networks can be categorized into four main types: Invariant GNNs: These networks, such as SchNet (Schütt et al., 2017), DimNet (Gasteiger et al., 2020), SphereNet (Liu et al., 2022), and GemNet (Gasteiger et al., 2021), encode the invariant geometric information like distances and directions directly into their model design to consider the 3D structures. Cartesian equivariant GNNs: Networks like GVP-GNN (Jing et al., 2020), PaiNN (Schütt et al., 2021), and E(n)GNN (Satorras et al., 2021) further consider direction vector as input and use scalar-vector operations to consider their interactions within the architectures. Spherical Equivariant GNNs: These networks such as TFN (Thomas et al., 2018), SEGNN (Brandstetter et al., 2021), NequIP (Batzner et al., 2022), Equiformer (Liao & Smidt, 2022), Allegro (Musaelian et al., 2023), MACE (Batatia et al., 2022), usually use the spherical harmonics of the directions as the input spherical tensors. Then they combine spherical tensors using equivariant operations like Tensor Product (TP) and convert them into irreducible representations. These networks have more complex interactions between equivariant irreducible representations, demonstrating superior performance and widespread application in property prediction (Ramakrishnan et al., 2014), force field prediction (Chmiela et al., 2017), and Hamiltonian matrix prediction (Schütt et al., 2019; Yu et al., 2024). Given the widespread use of the powerful spherical equivariant GNNs, understanding their key components, especially Tensor Product (TP), is one of the most essential problems in studying the

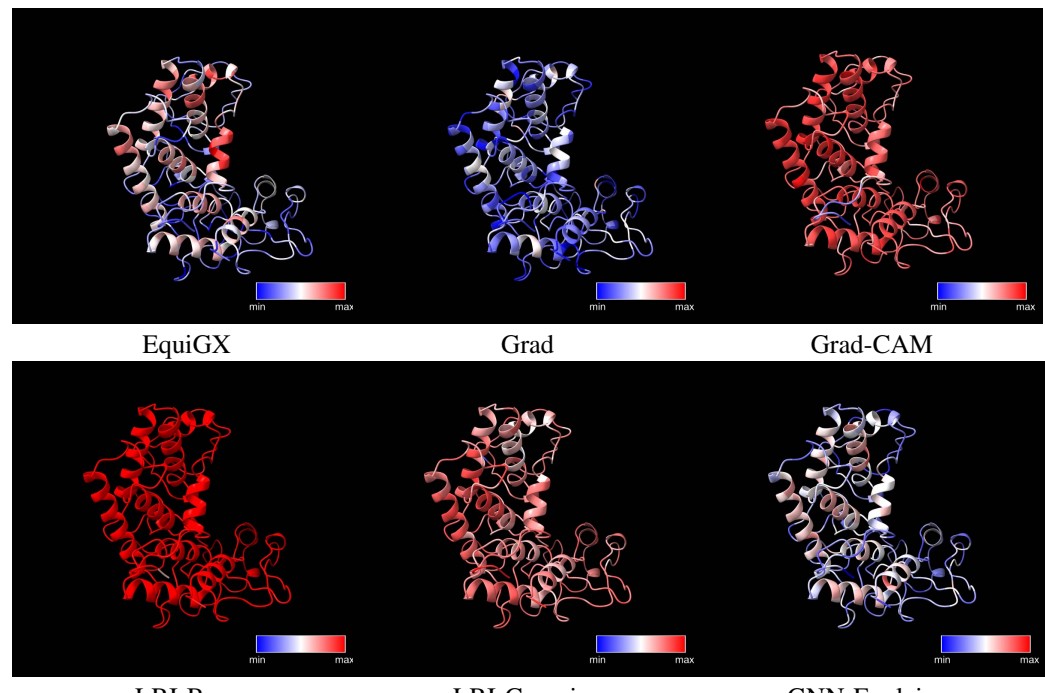

Figure 8: Explanation results on the SCOP dataset of all-alpha proteins. Since the sample is an all-alpha protein, ideally the $\alpha$-helices should have high importance scores, i.e. be red in the figure.

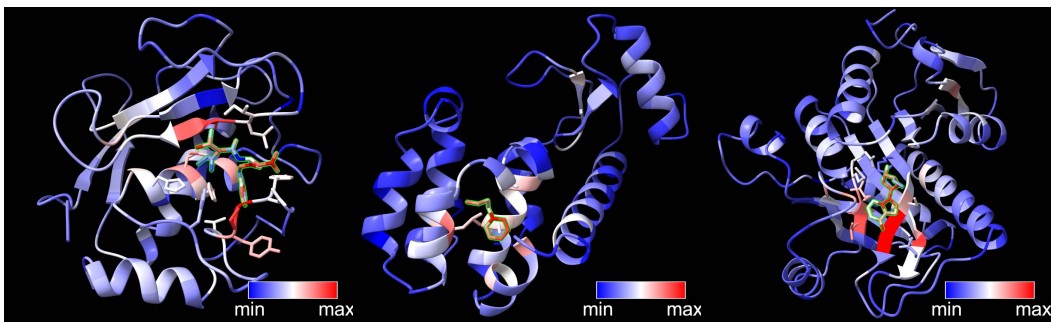

Figure 9: Explanation results of EquiGX on the BioLip dataset. The ligand is highlighted with a green border.

explainability of equivariant GNNs. While the previous three types of networks explicitly encode the invariant or equivariant symmetry within their networks, the networks in unconstrained GNNs (Hu et al., 2021) are not necessarily rotational invariant or equivariant for efficient training and inference. Furthermore, FAENet (Duval et al., 2023b) makes use of frame averaging techniques to make sure the overall framework maintains rotational invariant and equivariant.

