# OpenReview forum: "On Explaining Equivariant Graph Networks via Improved Relevance Propagation"
_ICLR.cc/2025/Conference — Submitted to ICLR 2025_

### Official Review · Reviewer_ZD9s · 2024-10-24

**Soundness:** 2
**Presentation:** 2
**Contribution:** 2
**Rating:** 3
**Confidence:** 2

**Summary:**

This paper proposed a novel method, known as EquiGX, which uses the Deep Taylor decomposition framework to extend the layer-wise relevance propagation rules tailored for spherical equivariant GNNs.

> **[First Comment, Rating 3, Confidence 3, with 2/2/2 of Soundness/Presentation/Contribution]**

I don't quite understand the meaning of this article, and I have doubts about several key formulas and experiments. I chose 3 conservatively rather than 5. If the author can convince me in the rebuttal, I am willing to increase the score.

**Strengths:**

Building an interpretable equivariant graph neural network plays a very important role in the actual implementation of AI for Science, and the research object of this article is very valuable. In addition, the visualization of the experimental results looks very interesting and worth further consideration.

**Weaknesses:**

>  **W1. Interpretability does not seem to be an invariant scalar.**

It seems that the interpretability of nodes should be an invariant scalar, but Hadamard division and Hadamard multiplication are used in the calculation process, which does not seem to be an invariant scalar? If this property is not an invariant scalar, the analysis of equivariant neural networks seems meaningless, and it seems that it is not as good as studying traditional graph neural networks.

> **W2. The practical significance of node interpretability.**

In actual scenarios, many interactions involve a subgraph, such as a benzene ring or a sulfonic acid group. Discussing the interpretability of a single node seems more like a toy and has no practical significance.

> **W3. Is EquiGX generalizable to other models?**

The article claims to "extend the layer-wise relevance propagation rules tailored for spherical equivariant GNNs". I am curious whether it can be extended to other common high-degree steerable models (e.g. SEGNN [a], Cormorant [b], NequIP [c], Allegro [d], MACE [e]). In addition, can the models using high-degree steerable features based on scalarization methods (e.g. SO3KRATES [f], HEGNN [g]) be explained? I am not asking the authors to discuss all of them. Can authors analyze one or two and make simple analogies for the rest?

> **W4. Will the symmetrical structure of the synthetic experiment cause potential problems?**

Some literature [h, g] points out that symmetric structures can cause the degradation of equivariant graph neural networks of a certain order. Will this affect the experimental results? Are there any potential errors?

> **W5. Explanation of Fig. 3-4.**

This graph looks very interesting. Does it have any specific meaning? Also, is this graph specially selected? Can other data results have such a sharp contrast?

> **W6. More baselines and dynamic datasets.**

Is it possible to add more baselines [a-g] to these experiments? And can EquiGX be tested on dynamic datasets (e.g. CMU Motion Capture dataset [i], Adk equilibrium trajectory dataset[j])?

> **W-Others. Typos and suggestions.**

- [Line 104] $\mathcal{C}\_{(\ell\_1, m\_1),(\ell\_2, m\_1)}^{(\ell\_3, m\_3)}$should be $\mathcal{C}\_{(\ell\_1, m\_1),(\ell\_2, m\_2)}^{(\ell\_3, m\_3)}$.
- [Line 212] $d\_ij$ should be $d\_{ij}$.
- It is recommended not to mix degree and order, as the latter is generally used to represent multi-body interactions. In addition, the Taylor decomposition in this article also has order, which can easily cause confusion.

[a] Cormorant: Covariant molecular neural networks.

[b] Geometric and physical quantities improve e(3) equivariant message passing.

[c] E(3)-equivariant graph neural networks for data-efficient and accurate interatomic potentials.

[d] Learning local equivariant representations for large-scale atomistic dynamics

[e] Mace: Higher order equivariant message passing neural networks for fast and accurate force fields.

[f] A Euclidean transformer for fast and stable machine learned force fields

[g] Are High-Degree Representations Really Unnecessary in Equivariant Graph Neural Networks?

[h] On the expressive power of geometric graph neural networks

[i] Carnegie-mellon motion capture database

[j] Molecular dynamics trajectory for benchmarking mdanalysis.

**Questions:**

See Weakness.

---

### Official Review · Reviewer_4FtR · 2024-10-31

**Soundness:** 3
**Presentation:** 3
**Contribution:** 2
**Rating:** 5
**Confidence:** 4

**Summary:**

This paper introduces an approach to explain the prediction of equivariant graph neural networks via relevance propagation by leveraging the first-order term in Taylor expansion. The score is demonstrated to be conservative through layers, and the authors have derived the scores for different operations including tensor products, linear layers and non-linearities such as norm-based non-linearity. Experiments are performed on two synthetic datasets and three real-world datasets, showing the efficacy of the proposed approach.

**Strengths:**

1. The method is well-motivated and easy to follow. It is an adaptation of layer-wise relevance propagation on the domain of equivariant graph neural networks.

2. Besides simple synthetic datasets, the method has also been benchmarked on large-scale protein datasets which should be appreciated.

3. The experiment results seem promising. The proposed method offers significant improvement over the existing approaches. The explanations have much more semantics.

**Weaknesses:**

1. The method seems to be a straightforward implementation of layer-wise relevance propagation.

2. Lacking experiments on more backbones that are widely adopted in equivariant graph neural networks literature. Please see Q1.

3. Some experiment settings lack sufficient justification compared with previous works and the results seem troublesome. Please see Q2.

**Questions:**

1. While the work primarily adopt TFN as the backbone, is it able to generalize to other equivariant GNNs, such as EGNN [1] and ClofNet [2]? Tensor-product based equivariant GNNs are intriguing but may have limited applicability in certain scenarios and more investigations on the scalarization-based approaches should be necessary. If the proposed approach is general for tensor-product based GNNs, it should also be able to be applied to scalarizaion-based methods with degree restricted to 0 and 1. Adding more experiments on these settings would enhance the contribution of the paper.

2. For the experiments on ActsTrack, the results (e.g., auroc) are much worse than those reported in the original work of LRI [3]. While I understand you use TFN instead of the backbones (e.g., EGNN) used by LRI, the inferior results pose a concern on why we even want to use TFN in these scenarios since it is more complicated, potentially more time-consuming, while more importantly, incurs worse performance.

That said, I am open to discussion and increasing the score if the concerns are properly addressed.

[1] Satorras et al. E(n)-equivariant Graph Neural Networks. ICML'21.

[2] Du et al. SE(3) Equivariant Graph Neural Networks with Complete Local Frames. ICML'22.

[3] Miao et al. Interpretable Geometric Deep Learning via Learnable Randomness Injection. ICLR'23.

---

### Official Review · Reviewer_5Fc2 · 2024-11-02

**Soundness:** 2
**Presentation:** 2
**Contribution:** 2
**Rating:** 3
**Confidence:** 4

**Summary:**

The paper introduces a novel method called EquiGX aimed at enhancing the explainability of equivariant graph neural networks (GNNs) specifically designed for 3D geometric graphs.

**Strengths:**

1. The proposed method, EquiGX, is specifically tailored for geometric graphs, addressing the unique challenges posed by 3D data structures that many existing XAI methods fail to tackle effectively
2.Empirical results indicate that EquiGX excels at identifying critical geometric properties when compared to alternative approaches. This is evidenced by significant improvements in quantitative metrics such as AUROC and average precision across various datasets.
3.he method introduces a novel decomposition of prediction scores that allows for detailed analysis of each layer's contribution to the final predictions. This insight is crucial for understanding model behavior in complex architectures.

**Weaknesses:**

1.
The paper lacks a robust foundation for its motivation, asserting that existing explanation approaches for 2D graphs are inadequate for 3D graphs without providing comprehensive discussion or evidence to support this claim. It fails to specify the characteristics of current methods detailed in Section 2.2 that render them unsuitable for geometric graphs, leaving unclear what specific aspects of these approaches are overly tailored to 2D structures. Additionally, the authors do not explore how these methods could be naively extended to 3D contexts and, more importantly, why such adaptations would be ineffective; including a clear example of a naïve extension along with an analysis of its shortcomings would significantly strengthen the argument.

2.The statement in lines 123-225, which claims that "despite significant advances in XAI for 2D GNNs, these methods primarily focus on evaluating the importance of edges, nodes, and subgraphs, struggling to incorporate positional information effectively and fully evaluate the importance of geometric features," lacks clarity and soundness. The authors should elaborate on the reasons behind this struggle; for instance, why can't coordinates be effectively treated as features, and what limitations do existing methods for highlighting features present? Additionally, the challenges associated with emphasizing edges and nodes require further explanation, especially since Section 4 later employs these very techniques to assess the method against existing approaches. Similarly, the subsequent assertion that applying these techniques to 3D geometric graphs, particularly in the context of equivariant GNNs, poses significant challenges is vague. It would greatly enhance the paper if the authors provided a detailed discussion of the specific challenges encountered in this area.

3. The paper lacks citations for several key feature-based explanation methods in neural networks, such as integrated gradients, which are also applicable to graph neural networks (GNNs), e.g. [1]. The authors should consider expanding the discussion to include interpretable GNNs in addition to focusing solely on post-hoc explanation techniques, e.g. [2].

[1] Axiomatic Attribution for Deep Networks, Sundararajan et al.,  2017.
[2]The Intelligible and Effective Graph Neural Additive Networks, Bechler-Speciher et al., NeurIPS 2024.

**Questions:**

1. What specific characteristics of the current explanation approaches for 2D graphs, as discussed in Section 2.2, do the authors believe render them inadequate for 3D geometric graphs, and how could these methods be naively extended to 3D contexts? Furthermore, what reasons can the authors provide to demonstrate why such adaptations would be ineffective?

2. What specific reasons can the authors provide to clarify why existing XAI methods for 2D GNNs struggle to incorporate positional information and fully evaluate geometric features, particularly regarding the treatment of coordinates as features?

3. What challenges are associated with emphasizing edges and nodes, especially in light of their later use in Section 4 to evaluate the proposed method against existing approaches?

---

### Official Review · Reviewer_vtsA · 2024-11-04

**Soundness:** 2
**Presentation:** 3
**Contribution:** 2
**Rating:** 6
**Confidence:** 4

**Summary:**

In this paper, the authors propose an explanation technique based on the equivariant graph networks. The idea is based on the Taylor decomposition to assign relevance scores to each part in the network. In addition, the authors discuss separately about how to compute the relevance score for different components in the network. Experimental results seem to show the effectiveness of the proposed method in quantitative and qualitative evaluations.

**Strengths:**

- The derivation of the method is easy to follow. The authors give sufficient preliminary knowledge about the basic knowledge of equivariant graph networks.

- Experiments show good visualization to demonstrate the effectiveness of the method.

**Weaknesses:**

- My concerns in experiments stem from the backbone model and datasets. It seems the authors use only the TFN to conduct experiments. Can the proposed method be applied to other geometric GNNs like EGNN or SE(3)-Transformer? If so, it is recommended to include these backbones to strengthen the experiments. I also found the recent baseline LRI, showing higher scores in the ActsTrack dataset from the original paper than it is reported in this paper. The deviation should be explained. In addition, compared with synthetic datasets, I am more interested in seeing the comparison in other datasets used in LRI but not included in this paper.

- I think the proposed method is reminiscent of previous gradient-based methods via propagation. In addition, the taylor decomposition also requires the computation of gradients. However, previous gradient-based methods are known to be sensitive to the model randomization [1]. I am concerned if this is also an issue to the proposed method given that the variance in table 3 is high in most datasets. Also, how this variance is computed is not mentioned in the paper.

[1] Adebayo, Julius, et al. "Sanity checks for saliency maps." Advances in neural information processing systems 31 (2018).

**Questions:**

See the weakness above

---

### Meta-Review · Area_Chair_gXVU · 2024-12-20

**Metareview:**

EquiGX extends layer-wise relevance propagation (LRP) for spherical equivariant GNNs using Deep Taylor decomposition, providing detailed insights into how geometric data affects predictions. Experiments show it outperforms baselines, offering improved interpretability for equivariant GNNs.

### Strengths:

1. The method is well-motivated and easy to follow.
2. Experiments provide clear visualizations that effectively demonstrate the method’s effectiveness.

### Weaknesses:

1. The paper lacks a strong theoretical foundation for its motivation.
2. The method appears to be a straightforward implementation of layer-wise relevance propagation.
3. The experiments do not include a broader range of widely-adopted backbones from the equivariant graph neural network literature.
4. Some experimental settings lack sufficient justification compared to prior work, and the results seem problematic.
5. The paper omits citations to several key feature-based explanation methods in neural networks.

### Overall:

The paper exhibits notable weaknesses in terms of novelty, significance, and clarity. Therefore, a rejection is recommended.

**Additional Comments On Reviewer Discussion:**

The authors did not provide any feedback in the rebuttal period.

---

### Decision · Program_Chairs · 2025-01-22

Reject